# The development of respect in young athletes: A systematic review and meta-analysis

**Manuel Tomás Abad Robles***, **Benjamín Navarro Domínguez, José Antonio Cerrada Nogales, Francisco Javier Giménez Fuentes-Guerra**

Facultad de Educación, Psicología y Ciencias del Deporte., Universidad de Huelva, Huelva, Spain

* manuel.abad@dempc.uhu.es

**Data Availability Statement:** All relevant data are within the manuscript and its Supporting information files.

## Abstract

### Background

The practice of sports can lead to the development of values such as respect, self-control, effort, autonomy and leadership. However, sport can only foster educational habits and values if it is properly timed and specifically focused towards that end. The aim of this study was to carry out a systematic review and meta-analysis of the effect of interventions on the development and promotion of respect in the practice of sport among young people.

### Methods

A systematic search was conducted, according to the guidelines of the PRISMA declaration, in the Web of Science (WOS), PubMed (Medline), Scopus, Google Scholar and SportDiscus databases. A total of 6 articles were considered to meet the inclusion criteria for the promotion of respect. Criteria for inclusion included: the full text needed to be available; it should be written in one of the selected languages: English, Spanish and Portuguese; it should be an intervention, an experimental or quasi-experimental study or a randomized controlled trial. Each manuscript was independently reviewed by three authors of this work.

### Results

The results of the meta-analyses showed that the Siedentop sports education model, and Hellison's personal and social responsibility model (TPRS), had significant improvements regarding respect for opponents (total effect size = 0.39, small effect, with 95% Confidence Interval of 0.07 to 0.72). In addition, these models, along with another programme called Fair Play, also achieved significant increases as regards respect for the social conventions of sport (total effect size = 0.67, moderate effect, with 95% IC from 0.25 to 1.10).

### Conclusions

In conclusion, the use of interventions related to the above-mentioned models is recommended when it is intended to encourage respect for opponents and social conventions.

**Funding:** The author(s) received no specific funding for this work.

**Competing interests:** The authors have declared that no competing interests exist.

These considerations could be useful to both teachers and coaches in order to further cultivate these important attitudes.

## Introduction

The practice of sports can lead to the development of values such as respect, self-control, effort, autonomy and leadership [1]. However, sport can only foster educational habits and values if it is properly timed and specifically focused towards that end [2–4]. The teacher or coach has a great responsibility, because their work involves the conveyance of a series of values which can be used in the field of sports and extrapolated to other contexts [5]. Thus, numerous authors agree on the need to vindicate and show the educational role of sport [2], as long as it is undertaken in a conscious, planned and coherent way, since sport in itself does not convey values or counter-values [6], but it is the teacher or coach who can assign a pedagogical value to the practice of sport [7].

In this sense, with the purpose of an adequate promotion and development of values within the practice of physical activity and sport, different proposals have appeared. On the one hand, Rockeach in 1973 based his theory of value hierarchy on the classification of values and attitudes in sports contexts [6], paying special attention to differentiate between instrumental or ambivalent values, which are those aimed at both good and bad causes or actions, and final or ethical values, differentiated in emotional identification and self-sacrifice. From the point of view of the teacher or coach as a role model, it arises the model set forth by Wandzilak [8]. According to this, the teacher or coach must be committed to the development of values using the ecological model. The first model of education in social responsibility was proposed by Hellison [9]. It was designed to help groups of young people at risk of social exclusion to have a positive experience which would help them build their personal and social skills, as well as their responsibilities, both in sport and in life [10]. Values associated with responsibility, effort and independence, along with values related to social responsibility entail respect for feelings and rights, empathy and social sensitivity [11]. Finally, another model which seeks to develop responsibility through sport practice is the sport education proposed by Siedentop [12] which aims to contribute to ethical development and fair play from the motor, attitudinal and cognitive fields.

Lee [13] considers that sportsmanship is based on the principle of justice for all, where there is no intentionality (accidental or deliberate) to achieve a dishonest advantage over the rival. Therefore, sportsmanship refers to behaviour defined by "positive interaction with teammates, referees, coaches and opponents" [14]. Studies on sportsmanship, both in physical education lessons and in sports schools, have demonstrated that sportsmanship can prevent violent behavior [15] and encourage pro-social behaviour [16]. Furthermore, from a social and psychological perspective, Vallerand et al. [17] found that sportsmanship involves five factors: 1) respect for social conventions; 2) respect for rules and referees; 3) respect for opponents; 4) commitment, effort and total commitment to their sport; 5) negative approach to the sport or disruptive player behaviour. Accordingly, fair play and sportsmanship synthesise the code of ethics in sport and are related to individual and social ethics. Thus, the values integrated by social ethics make reference to respect for the rules of the game, opponents and the referee [18], and to the fundamental principles of justice [19]. Pro-social behaviours such as respect can have positive effects on people's social-emotional development, facilitating an emotional identification of one another [6], which supports the implementation of intervention

programmes as strategies to prevent antisocial behaviours. Along these lines, Brackenridge et al. [20] developed the "respect" programme, supported with funds from the English football federation and aimed at coaches, governing bodies, players and spectators.

Previous researches have analysed respect from different perspectives. Hassandra et al. [21] applied a methodology based on Olympic education in order to study respect for teammates. Lamoneda et al. [22, 23] undertook interventions focused on improving the personal and social aspects of sportsmanship. In turn, Malinauskas and Juodsnukis [24] examined respect for others using the personal and social responsibility model, while Pan et al. [25] investigated the influence of using a hybrid model between the personal and social responsibility model and the sports education model on respect for class rules and others.

Respect, in other words, can be developed as a personal value in terms of one's social image or as a social value in which respect for others prevails [26], which can ensure an ability to tolerate the behaviour of others in sporting contexts [27]. It should also be noted that physical activity and sport become an appropriate and ideal context for the development of values such as respect [28]. In this way, children who participate in sport can benefit from the development of a range of personal and social skills, such as skills for relationships with peers and for personal and social responsibility, as well as pro-social behaviours such as respect [7]. Therefore, it is pertinent to study the development of respect in young people through sport practice.

As far as we know, the study of the effects of interventions focused on the development of respect is novel. For this reason, and considering the importance of the development of personal and social responsibility, and respect as a value associated with the implementation of good behaviours in the practice of physical activity and sport, the aim of this study was to conduct a systematic review and meta-analysis of the effect of interventions on the development and promotion of respect within the practice of sport among young people. In this regard, the review and meta-analysis questions are as follows: 1. Are interventions aimed at developing the respect of boys and girls during sport effective? 2. What are the characteristics of the interventions that are most conducive to the development of respect?

## Methods

This study was performed by means of a systematic review and meta-analysis in which existing research on the development of respect in the sports and physical education field was studied and analysed. To undertake this systematic review, the PRISMA declaration and the practical guide to systematic reviews with or without meta-analysis were used [29, 30].

### Inclusion criteria

The inclusion criteria applied in this study were: a) the full text needed to be available; b) it should be written in one of the selected languages: English, Spanish and Portuguese; c) it should be an intervention, an experimental or quasi-experimental study or a randomized controlled trial. Consequently, the manuscripts were included as a result of screening based on the different eligibility criteria outlined above. Additionally, studies from other sources (analysis of the reference lists of selected articles) meeting the same inclusion criteria were incorporated.

### Search strategy

A systematic review was conducted following PRISMA guidelines [29, 30]. The search of the different articles was accomplished in five databases (Web of Science, Scopus, SportDiscus, Google Scholar and Pubmed) from September to 12 November 2020. Three differentiated search blocks were established for this purpose: 1) sport OR physical education; 2) respect; and

3) intervention OR experimental OR quasi-experimental OR randomised controlled trial. Once the search was completed, duplicate articles were eliminated and the search was narrowed down to articles published between 2000 and 2020.

## Selection of studies and data mining process

After completing the search of the different articles, the title and the abstract were analysed in order to find the most relevant ones and to exclude those which did not meet the inclusion criteria. In this way, six articles were selected to be used for data mining with the topic "respect" as the main subject matter to be taken into account in the different investigations. It should be noted that one of the articles envisaged two experimental groups. In order to reduce selection bias, each manuscript was independently reviewed by three authors of this work, who decided whether or not a paper met the inclusion criteria. In case there was no consensus on the inclusion or not of any study, the dilemma was solved by consulting the fourth author.

## Assessment of risk of bias

To assess risk of bias, the PEDro scale was used [31]. This scale was designed to analyse the quality of study interventions, mainly randomised controlled trials. Moreover, the grading of recommendations assessment, development, and evaluation (GRADE) approach was used to assess the quality of evidence [32]. This approach involves a four-point scale ("very low," "low," "moderate," and "high") and the quality of evidence decreases when there is inconsistency, poor applicability of evidence, inaccuracy, and publication bias. Table 1 shows the results of the risk of bias of the included manuscripts. Finally, to assess the risk of bias and the quality of evidence, each study was independently reviewed by two authors of this paper, who decided whether or not a paper met the inclusion criteria. In case there was no consensus on the inclusion or not of any study, the dilemma was solved by consulting the other researchers.

## Data collection

Firstly, data were obtained from the included articles. Later, the information gathered was verified by all the researchers. Following the recommendations of the PRISMA guidelines, relevant information included participants, intervention, comparisons, outcomes, and study design

**Table 1. Risk of bias according to the PEDro scale.**

| Study | Response to each item level of evidence | | | | | | | | | | | |
| --- | --- | --- | --- | --- | --- | --- | --- | --- | --- | --- | --- | --- |
| | 1 | 2 | 3 | 4 | 5 | 6 | 7 | 8 | 9 | 10 | 11 | Total |
| Burgueño and Medina-Casaubón [33] | Y | N | Y | Y | Y | N | N | Y | Y | Y | Y | 7 |
| Hassandra et al. [21] | Y | Y | N | Y | Y | N | Y | Y | Y | Y | Y | 8 |
| Lamoneda et al. [23] | Y | N | Y | Y | Y | N | N | Y | Y | Y | Y | 7 |
| Méndez-Giménez, Fernández-Ríos and Méndez-Alonso [34] | Y | N | Y | Y | Y | N | N | Y | Y | Y | Y | 7 |
| Sánchez-Alcaraz et al. [35] | Y | N | Y | Y | Y | N | N | Y | Y | Y | Y | 7 |
| Viciana et al. [36] | Y | N | Y | Y | Y | N | N | Y | Y | Y | Y | 7 |

Y: criterion fulfilled; N: criterion not fulfilled; 1: eligibility criteria were defined; 2: the participants were randomly distributed to groups; 3: the assigned was concealed; 4: the groups were similar before of the intervention (at baseline); 5: all participants were blinded; 6: therapists (teachers) who conducted the intervention were blinded; 7: there was blinding of all evaluators; 8: the measures of at least one of the fundamental outcomes were attained from more than 85% of the participants initially; 9: "intention to treat" analysis was conducted on all participants who received the control condition or treatment as assigned; 10: the findings of statistical comparisons between groups were reported for at least one fundamental outcome; 11: the study gives variability and punctual measures for at least one fundamental outcome; total score: each satisfied item (except the first) adds 1 point to the total score.

**Table 2. Characteristics of the participants, duration, instrument and protocol of the interventions included in the systematic review and meta-analysis.**

| Studies | Country | Sample size of groups and sex | Age (SD) and educational level/context | Duration of the study | Tools | Protocol | |
|---|---|---|---|---|---|---|---|
| | | | | | | Control Group | Experimental Group |
| Burgueño and Medina-Casaubón [33] | Spain | 148 (70 boys and 78 girls) | 16–18 years old | 8 weeks (16 sessions of 60 minutes. Two sessions per week) | The Spanish version of the multidimensional sportsmanship orientation scale (MSOS) [38] | Traditional teaching model | Methodology based on the sports education model [39, 40] |
| | | Control G. (74) | M = 17.04 (0.99) | | | | |
| | | Experimental G. (74) | Secondary education | | | | |
| Hassandra et al. [21] | Greece | 126 (71 boys and 55 girls) | 10–12 (NR*) | 10 one-hour sessions per week | The fair play questionnaire based on multidimensional sports-person-ship orientations Scale [41] | Traditional model (standard Olympic education curriculum) | Fair play programme included in the Olympic education programme [42] |
| | | Control G. (60) | Primary education | | | | |
| | | Experimental G. (66) | | | | | |
| Lamoneda et al. [23] | Spain | 126 (124 boys and 2 girls) | 12-Oct | 2 years 6 weeks of intervention, 20 minutes per week | MSOS. Spanish version adapted to U-12 football [43] | No approach was applied | Personal and social responsibility programme [9] |
| | | Control G. (65) | M = 10.99 (0.67) | | | | |
| | | Experimental G. (61) | Football club | | | | |
| Méndez-Giménez et al. [34] | Spain | 295 (159 boys and 136 girls) Control G. (110) | 12–17 years old | 12 sessions of 55 minutes | Spanish version of the MSOS by Vallerand et al. [41], validated in Spanish by Martín-Albo et al. [38] | Traditional model following a structure of skill teaching, exercises and games | Sports education model with conventional material (SE-CM); and model of sports education with self-built material (SE-AM) [39, 40] |
| | | Experimental G. Sports | M = 14.2 (1.68) | | | | |
| | | Education with conventional material (SE-CM) (107) and with alternative material (SE-AM) (78) | Secondary education | | | | |
| Sánchez-Alcaraz et al. [35] | Spain | 563 (323 boys and 240 girls) Control G. (292) | 12–15 years old | 4 months (2 hours of physical education per week) | The Spanish version of the MSOS [38] | The teacher's usual methodology | Personal and social responsibility model [9] |
| | | Experimental G. (281) (data provided by the authors) | M = 13.73 (1.83) | | | | |
| | | | Secondary education | | | | |
| Viciana et al. [36] | Spain | 123 (60 boys and 63 girls) | 14–15 years old | 12 sessions | The Spanish version of the MSOS [38] | Traditional approach of sport in physical education | Sport education model [39, 40] |
| | | Only 109 participants | (NR*) | | | | |
| | | Control G. (42) | Secondary education | | | | |
| | | Experimental G. (67) | | | | | |

(PICOS) [37]. Table 2 shows the characteristics of the participants, duration, instrument and protocol of the interventions.

Regarding the interventions, in the Burgueño and Medina-Casaubón study [33] the control group conducted a traditional approach in which the first 12 sessions focused specifically on the basic technical skills of basketball and its basic tactical elements. The last four sessions focused on competition, and the teacher checked the degree of compliance with the rules of the game. As for the experimental group, the intervention was based on the sports education model [39] in which three phases were established: The initial phase included an introductory session and a practice session led by the teacher. The autonomous practice phase had the purpose of developing the technical and tactical skills by combining team practice and competition. Lastly, the final phase included the regular competition and a culminating event with an awards ceremony included. In the study by Hassandra et al. [21], the control group followed the standard curriculum of the Olympic education programme [42]. In the experimental

group, the teacher received a four-day seminar on Olympic education, which included a five-hour session on social education in primary schools. The fair play programme was incorporated into the Olympic education programme in order to develop fair play behaviours. In the study by Lamoneda et al. [23], the control group followed their usual methodology without any intervention or moral treatment in the field of values education, while in the experimental group the values education programme was implemented. In this programme, the athlete was guided to acquire a commitment; to follow up on it; and to value the achievements. In order to establish the sequencing in the conveyance of values, Hellison's social and personal responsibility programme [9] was taken as a reference. The programme included four phases: 1) Meeting in the changing room: analysis of one's own behaviours; 2) Personal reflection and commitment; 3) Action: putting into practice the commitments made; 4) Reflection: review of commitments and reinforcement of behaviour.

On the other hand, in the study by Méndez-Giménez et al. [34] the control group followed the traditional model structured in: skill teaching, exercises and games. The lessons were led by the teacher and divided into three parts: the warm-up, skill improvement through exercises and final match. It should be mentioned that in the study by Méndez-Giménez et al. [34] the intervention was based on the sports education model [39] with three different phases: The initial phase involved an introductory session and a teacher-led practice in which teams were set up and the different roles were assigned. The autonomous practice phase aimed to develop the technical and tactical skills for which team practice and competition were combined. Lastly, the final phase included the regular competition and a culminating event with an awards ceremony included. It should be highlighted in this research that there were two experimental groups carrying out an intervention based on the sports education model with different materials. One group with conventional material, and another one with self-built material. In the first group, in order to promote membership, the students were part of a mixed and heterogeneous team which remained unchanged throughout the season. A competition calendar was organised from the beginning of the unit. On top of that, a final inter-class championship was held as a *grand finalé*. While, in the second group, the only difference was the use of self-built material instead of conventional ones, which were shared with players of the team itself and those of others.

Regarding the study by Sánchez-Alcaraz et al. [35], the students in the control group followed a usual methodology, being the session structured in a warm-up, a core part and a warm-down, while those in the experimental group followed an intervention based on the teaching model of personal and social responsibility. The daily format of the session was structured in four parts: awareness talk, responsibility, group meetings, assessment and self-assessment [10]. The teachers' performance was based on the implementation of the physical education session according to the standards of Hellison's personal and social responsibility teaching model, prioritising one of the levels of responsibility. Finally, in the study by Viciana et al. [36] the control group followed a traditional approach to sport in physical education with the following structure: brief explanation of the objectives of the session, a warm-up; then the core part of the session was developed, consisting of three to five tasks focusing on technique and small games which were changing the rules; and finally, a warm-down phase was applied at the end of the session. The experimental group followed the sports education model [39]. The intervention was divided into four phases: a) introductory phase during which the sports education programme was explained, the introductory games were played, the teams were distributed in a balanced way and roles were assigned; b) Pre-season: Autonomous practices were undertaken by teams and teachers, they supervised the classes, but left the practice of games and roles to the teams themselves; c) Season, where the formal competition between teams

took place; and, last but not least, d) a final event where the award ceremony was held as well as a final match between the participating teams.

## Statistical analysis

In the meta-analyses, a random-effects model was used to measure the effect of interventions regarding respect for opponents. The effect size was calculated using means and standard deviations before and after the appliance of the approach [44]. For this meta-analysis, the magnitude of Cohen's d was specified as follows: a) "large," for values greater than 0.8; b) "moderate," for scores between 0.5 and 0.8; c) and "small," for values between 0 and 0.5. Heterogeneity was assessed by calculating the following statistics: a) $Tau^2$, for the calculation of variance between studies; b) $Chi^2$; and c) $I^2$, which is a transformation of the $H$ statistic used to determine the percentage of variation caused by heterogeneity. The most common classification of $I^2$ considers values above 50% as highly heterogeneous, values between 25% and 50% as moderate and values below 25% as small [45]. The Review Manager 5.3 tool was used to perform all analyses [46].

## Results

### Selection of studies

After the initial search, 511 results were obtained: 203 from the SportDiscus database, 102 from Web of Science (WOS), 119 from Scopus, 75 from Pubmed (Medline) and 1 from Google Scholar. The documents were analysed and 11 additional studies were identified in the references of the documents or by other sources. Furthermore, duplicate articles were discarded. This resulted in the exclusion of 26 studies. Out of the remaining studies, 96 were excluded because they were systematic reviews or analyses of the literature and 383, after applying the exclusion criteria. Finally, after conducting the analysis, 6 studies were included in the meta-analysis, having met the different inclusion criteria established (see Fig 1). It should be highlighted that, although five of the six researches had been carried out in Spain, the intention of the researchers was to include the studies carried out in any country.

### Risk of bias

Table 1 shows the risk of bias of the four items included in the meta-analysis according to the PEDro Scale. The total scores of the manuscripts ranged from seven to eight points. As for the quality of evidence, the GRADE guidelines were followed. The quality of evidence was lowered on one occasion, due to the high degree of heterogeneity. As a result, the quality of evidence, according to GRADE's guidelines, was "very low," which was defined as "we have very little confidence in the estimate of the effect: the true effect is likely to be substantially different from the estimate of the effect" [32]. The assessment of the quality of evidence was conducted independently by two authors of this research, who decided whether or not a document met the inclusion criteria. In case there was no consensus on the inclusion or not of any study, the dilemma was solved by consulting the other researchers of this study.

### Characteristics of the studies

Table 2 displays a summary of the characteristics of the studies. There was a total of 1377 participants, out of which 734 were assigned to the experimental groups and 643 to the control groups. Out of the six researches included in the quantitative analysis, four were performed in secondary education with boys and girls between 12 and 18 years old, one in primary education (10–12 years old) and one in a football club (10–12 years old).

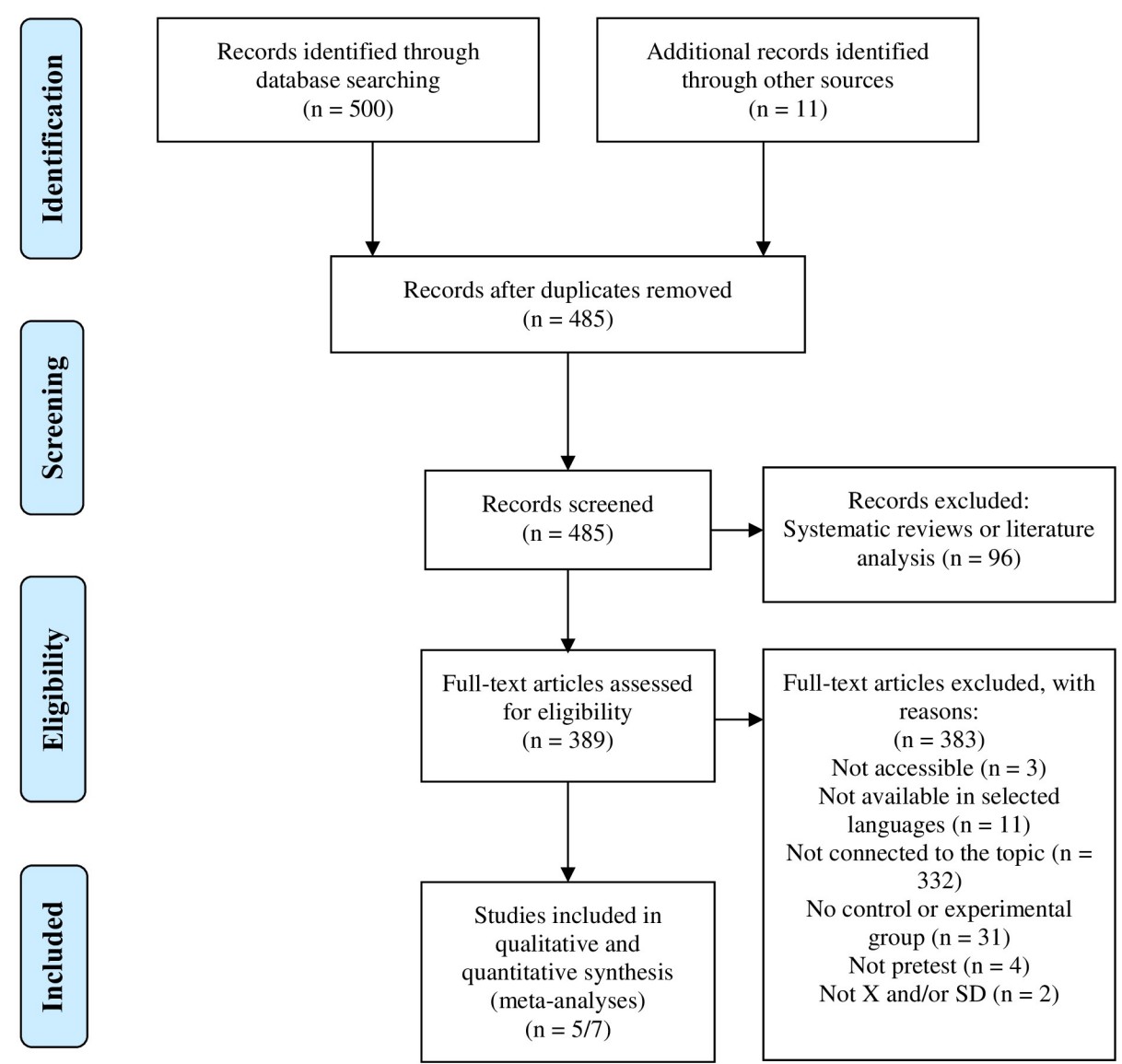

**Fig 1. Flow diagram for the systematic review process according to PRISMA statements.**

## Interventions

As shown in Table 2, the interventions of the experimental groups were based on the sports education model [33–34, 36], the personal and social responsibility teaching model [23, 35], and in a fair play programme [21], while the control groups followed the traditional teaching model or were not subjected to any approach at all. It is worth mentioning that the research by Méndez-Giménez et al. [34] included two experimental groups.

## Assessment of results

Figs 2–4 show the results of the variables studied (respect for opponents, respect for the social conventions of sport, and respect for rules and referees). In order to assess the respect for opponents, four studies [33–36] used the Spanish version of the MSOS by Vallerand et al. [41]

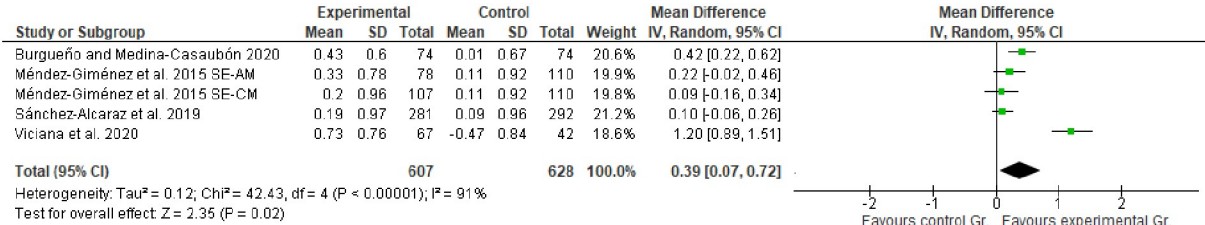

**Fig 2. Results of the meta-analysis regarding the effects of interventions on the respect for the opponent.**

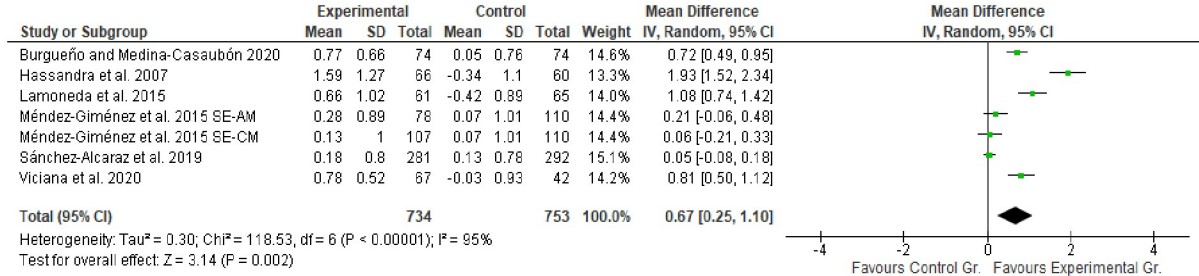

**Fig 3. Results of the meta-analysis regarding the effects of interventions on the respect for social conventions.**

validated in Spanish by Martín-Albo et al. [38]. This scale is made up of 25 items broken down into five subscales of sportsmanship: concern and respect for one's total commitment to participation in sport, respect for social conventions in sport, concern and respect for rules and referees, concern and respect for the opponent, and a negative approach to participation in sport. Furthermore, Lamoneda et al. [23] also used the Vallerand et al. Scale [41], but validated by Lamoneda et al. [43] for U-12 football. Hassandra et al. [21] used a Fair Play questionnaire [47] based on the Vallerand et al. Scale [41].

Fig 2 reflects that all five studies had improvements in the development of respect for opponents in participating subjects compared to subjects in the control groups, and were significant in two studies [33, 36]. However, no significant improvements were observed in the research of Méndez-Giménez et al. [34] (ED-MA), Méndez-Giménez et al. [34] (ED-MC), and Sánchez-Alcaraz et al. [35]. The total effect size was 0.39, with 95% CI from 0.07 to 0.72, which, according to the proposed classification, was a small effect size. However, the level of heterogeneity was large: $Tau^2 = 0.12$; $Chi^2 = 42.43$, $df = 4$ (p < 0.00001); $I^2 = 91\%$; test for total effect: $Z = 2.35$ (p = 0.02).

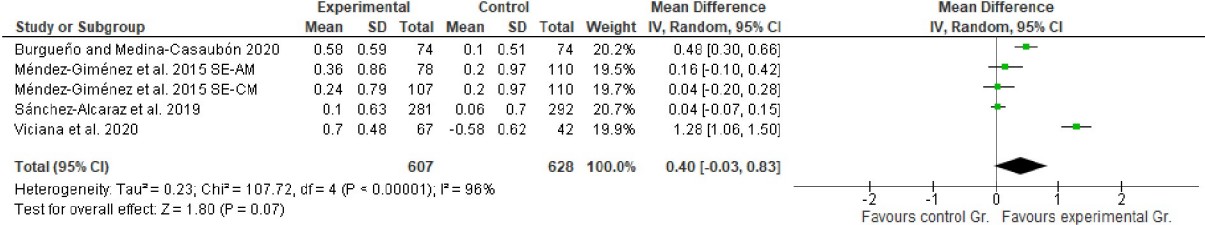

**Fig 4. Results of the meta-analysis regarding the effects of interventions on the respect for rules and referees.**

Fig 3 illustrates the effect of research interventions on the development of respect for the social conventions of sport in participating subjects. The results of the meta-analysis showed that all six studies had improvements in the development of respect for social conventions in comparison with subjects in the control groups. Out of the six studies, four demonstrated significant improvements [21, 23, 33, 36]. However, no significant improvements were found in the research of Méndez-Giménez et al. [34] (ED-MA), Méndez-Giménez et al. [34]. (ED-MC), and Sánchez-Alcaraz et al. [35]. The total effect size was 0.67, with 95% CI from 0.25 to 1.10. Following the proposed classification, the effect size was moderate. However, the level of heterogeneity was high: $Tau^2$ = 0.30; $Chi^2$ = 118.53, $df$ = 6 ($p < 0.00001$); $I^2$ = 95%; test for total effect: $Z$ = 3.14 ($p$ = 0.002).

Fig 4 presents the effect of research interventions on the development of respect for rules and referees. The results of the meta-analysis showed that all six studies obtained improvements in the development of respect for rules and referees in comparison with the subjects in the control groups. Out of the six studies, two showed significant improvements [33, 36]. However, no significant improvements were found in the research of Hassandra et al. [21]; Lamoneda et al. [23]; Méndez-Giménez et al. [34] (ED-MA); Méndez-Giménez et al. [34] (ED-MC); and Sánchez-Alcaraz et al. [35]. The total effect size was 0.40, with 95% CI from -0.03 to 0.83. Following the proposed classification, the effect size was small but not significant. Furthermore, the level 13 of heterogeneity was high: $Tau^2$ = 0.23; $Chi^2$ = 107.72, $df$ = 4 ($p < 0.00001$); $I^2$ = 96%; test for total effect: $Z$ = 1.80 ($p$ = 0.07).

## Discussion

The aim of this study was to conduct a systematic review and meta-analysis on the effect of interventions on the development and promotion of respect in the practice of sport among young people. Thus, the results of the meta-analysis showed that all six studies obtained improvements in the development of respect in comparison with the subjects of the control groups. Nevertheless, regarding the respect for opponents, the improvement was significant, but small, according to the size of the overall effect (0.39, with 95% CI from 0.07 to 0.72 with $p$-value = 0.02). Thus, the research by Viciana et al. [36] obtained a large effect size (1.20, with 95% CI from 0.89 to 1.51) in favour of the experimental group (see Fig 2). Additionally, the Burgueño and Medina-Casaubón study [33] also revealed outstanding results in favour of the experimental group (0.42, with 95% CI from 0.22 to 0.62. See Fig 2). It should be pointed out that both studies were performed using the Siedentop model of sports education [39], implemented in secondary education, with a length of 12 to 16 sessions of approximately 60 minutes each, and for the practice of volleyball and basketball. This way, it can be considered that interventions similar to those performed by Burgueño and Medina-Casaubón [33], and Viciana et al., [36], can achieve a significant contribution in the development of respect for opponents. The considerations in this meta-analysis may be useful for teachers and coaches to improve respect for opponents.

In terms of respect for the social conventions of sport, Fig 3 shows a significant moderate increase over the total effect size (0.67, with 95% CI from 0.25 to 1.10 with p-value = 0.002). Nonetheless, due to the high heterogeneity and low quality of the evidence, interpretation of the results of these meta-analyses should be considered carefully. On the other hand, in terms of respect for rules and referees, no significant improvement was found (0.40, with 95% CI from -0.03 to 0.83 with p-value = 0.07). In addition, it is worth noting that, out of the six studies, four reported significant improvements [21, 23, 33, 36] (see Fig 3), which showed the effectiveness of the three types of interventions used (Hellison's personal and social responsibility teaching model, sports education, and the fair play programme). The research by Hassandra

et al. [21] based on a fair play programme, which obtained a large effect size (1.93, with 95% CI from 1.52 to 2.34), is worth highlighting. Once again, the sports education model obtained important results in two studies [33, 36], which is relevant because through this model the student becomes an expert and a competent player who understands and values sport, being able to distinguish between good and bad behaviours in the practice of sport [39]. Thus, the considerations in this meta-analysis may be useful for teachers and coaches to improve respect for the social conventions of sport.

Concerning the respect for rules and referees, the research of Viciana et al. [36] obtained a large effect size (1.28, with 95% CI from 1.06 to 1.50) in favour of the experimental group. In addition, the study by Burgueño and Medina-Casaubón [33] also showed outstanding results in favour of the experimental group (0.48, with 95% CI from 0.30 to 0.66. See Fig 4). It is worth mentioning that both studies used the Siedentop sports education model [12]. Along these lines, Lamoneda et al. [22, 23], using a values education programme based on moral development and the personal and social factors of sportsmanship, did not obtain improvements regarding the respect for the rules and not cheating, but did obtain improvements concerning the respect for the referee when he or she makes a mistake, respectively. Meanwhile, Pan et al. [25] conducted an intervention using a hybrid model between the personal and social responsibility model and the sports education model, although they also found no significant improvements in terms of following classroom rules. Due to the results obtained, both in this research and in previous studies, it seems necessary to further research on intervention protocols which focus on improving respect for rules and referees.

It should be stressed that the most widely used instrument was the MSOS by Vallerand et al. [41], either in its Spanish version by Martín-Albo et al. [38] or the Spanish version adapted to U-12 football by Lamoneda et al. [43]. This instrument was used in five of the six articles analysed and shows adequate levels of reliability and validity [43]. The only study not using the MSOS was that of Hassandra et al. [21], who used a fair play questionnaire [47] based, however, on the Vallerand et al. as it can be observed, all the studies used practically the same instrument. As for the intervention protocols, no specific intervention was applied to the control groups, since in all the studies the traditional model was followed or no approach was applied. On the other hand, it should be highlighted that the most widely used intervention was based on the sports education model [33, 34, 36], followed by Hellison's personal and social responsibility teaching model [23, 35], while Hassandra et al. [21] followed a fair play programme.

Finally, to answer the questions of the review and meta-analysis raised above, we have to say that, overall, the interventions carried out in the studies analyzed significantly promoted respect for opponents and social conventions. In terms of the characteristics of the interventions that led to an improvement in respect, the results showed that interventions based on models of sports education and personal and social responsibility achieved a significant increase in respect.

To the best of our knowledge, this is the first systematic review with meta-analysis focused on examining the effects of interventions on the development of respect in physical education classes and sports clubs, using a rigorous and widely accepted methodology (PRISMA), and providing conclusions based on existing evidence. While the results found point to the recommendation of implementing the sport education model to promote respect, it should be noted that further studies are needed to increase the quality of evidence. Future research should deepen the study of the effects produced by the models of sports education [12], personal and social responsibility [9], fair play programme [42], and other methodologies and interventions related to the promotion of values such as respect, from all perspectives (peers, opponents, rules, referees, audience, material. . .), in boys and girls of different ages, and in both

educational and sports contexts. In addition, in order to determine the most appropriate context for the implementation of a particular model, these models could be applied in different situations (primary education, secondary education, sports schools, sports clubs, etc.). This would allow us to test their effectiveness in developing respect, and to analyse in which contexts they work best.

Nonetheless, the systematic review with meta-analysis conducted has some limitations. One of the main ones is related to the scarcity of articles found which met the established inclusion criteria in order to be selected for inclusion in the meta-analysis, which indicates the need for further research. Another limitation relates to the fact that the search for documents was limited to three languages: Spanish, English and Portuguese. Therefore, the risk of excluding manuscripts written in other languages could be high. Finally, it should be noticed that the data of the meta-analysis showed a high level of heterogeneity, which means that the interpretation of the results of this study should be considered carefully.

## Conclusions

Based on the study conducted, it can be concluded that the development of respect among young athletes can be successfully achieved through a planned and deliberate intervention, mainly in physical education classes. Thus, the first conclusion of the research is that the application of the model of sport education was shown to be an approach with great capacity to promote and develop respect for opponents and for the social conventions of sport while practicing sport. On the other hand, the personal and social responsibility model and the fair play programme can also lead to significant improvements in the development of respect. Finally, it seems that a traditional methodology fails to encourage and increase respect. These considerations may be useful and of practical application to teachers and coaches when seeking to develop respect for opponents and the social conventions of sport in their students and players. Nevertheless, these conclusions should be considered carefully, given the high heterogeneity and low quality of the evidence.

## Supporting information

**S1 Checklist. PRISMA 2009 checklist.**
(DOC)

## Author Contributions

**Conceptualization:** Manuel Tomás Abad Robles, Francisco Javier Giménez Fuentes-Guerra.

**Data curation:** Manuel Tomás Abad Robles, Benjamín Navarro Domínguez, José Antonio Cerrada Nogales.

**Formal analysis:** Manuel Tomás Abad Robles, Benjamín Navarro Domínguez, José Antonio Cerrada Nogales.

**Investigation:** Manuel Tomás Abad Robles, Francisco Javier Giménez Fuentes-Guerra.

**Methodology:** Manuel Tomás Abad Robles, Benjamín Navarro Domínguez, José Antonio Cerrada Nogales, Francisco Javier Giménez Fuentes-Guerra.

**Resources:** Benjamín Navarro Domínguez, José Antonio Cerrada Nogales.

**Supervision:** Manuel Tomás Abad Robles, Francisco Javier Giménez Fuentes-Guerra.

**Writing – original draft:** Manuel Tomás Abad Robles, Benjamín Navarro Domínguez, José Antonio Cerrada Nogales, Francisco Javier Giménez Fuentes-Guerra.

**Writing – review & editing:** Manuel Tomás Abad Robles, Francisco Javier Giménez Fuentes-Guerra.

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
