## [Decision Letter · Decision Letter 0]

21 Apr 2021

PONE-D-21-05952

The development of respect in young athletes: A systematic review and meta-analysis

PLOS ONE

Dear Dr. Abad Robles,

Thank you for submitting your manuscript to PLOS ONE. After careful consideration, we feel that it has merit but does not fully meet PLOS ONE’s publication criteria as it currently stands. Therefore, we invite you to submit a revised version of the manuscript that addresses the points raised during the review process.

We look forward to receiving your revised manuscript.

Kind regards,

Francisco Javier Huertas-Delgado, Ph.D.

Academic Editor

PLOS ONE

Journal Requirements:

Please provide the full electronic search strategy for at least one database, including any limits used, such that it could be repeated

Please include captions for your Supporting Information files at the end of your manuscript, and update any in-text citations to match accordingly. Please see our Supporting Information guidelines for more information: http://journals.plos.org/plosone/s/supporting-information.

Reviewers' comments:

Reviewer's Responses to Questions

**Comments to the Author**

1. Is the manuscript technically sound, and do the data support the conclusions?

Reviewer #1: Yes

Reviewer #2: Yes

Reviewer #3: Partly

Reviewer #4: Yes

2. Has the statistical analysis been performed appropriately and rigorously? 

Reviewer #1: Yes

Reviewer #2: Yes

Reviewer #3: Yes

Reviewer #4: Yes

3. Have the authors made all data underlying the findings in their manuscript fully available?

Reviewer #1: Yes

Reviewer #2: Yes

Reviewer #3: Yes

Reviewer #4: Yes

4. Is the manuscript presented in an intelligible fashion and written in standard English?

Reviewer #1: Yes

Reviewer #2: Yes

Reviewer #3: Yes

Reviewer #4: Yes

5. Review Comments to the Author

Reviewer #1: The development of respect in young athletes: A systematic review and meta-analysis

First of all, the reviewer would like to thank the authors for their work and efforts in trying to improve sports science knowledge.

General comments to the authors

Overall, this is a nice study that could have well systematic evaluation when integrated with the practice of sports among young people relating he development of values. The authors are commended on their efforts thus far. The study is well well-written and a great systematic review and meta-analysis. However, I suggest only small corrections to the authors and these corrections will allow improving the manuscript.

Abstract

Line 27: total effect size should be written with its’ magnitudes

Line 29: total effect size should be written with its’ magnitudes

Introduction section

Line 65: the authors should delete (p. 15)

Methods section

Line 118: the authors should rewrite sentence especially dates

Line 186: the authors should be attention to write capital letter throughout the article

Line 270: the authors should delete (p. 404)

Results section

This section is well designed and well-written.

Discussion section

Overall the discussion is well-written and incorporates relevant literature.

Overall the discussion section should be re-design according to journal guidelines especially margins

References

The authors should check writing style of references according to journal guidelines

Reviewer #2: The topic of the article seems interesting especially as the authors try to present 6 papers of systematic review and a meta-analysis on the development of respect for young athletes. Introduction - the authors present current research on the topic of this article, also highlight certain research that is focused on the direction of the study. I think that a phrase regarding the novelty of the study would be necessary in this chapter to be introduced

Line 118 - ''articles published between 2000 16 and 2020'', I think it's a mistake of word processing, please correct it....is 2000 - 2020, or 2000, 2016, 2020???

Another question for the authors refers to the fact that it has been highlighted that most of the studies are from Spain. So, do you have articles from other countries or did you want to analyze especially the research in this country? it is quite strange that 5 articles are from Spain and only one from another country, please detail because this aspect is important and I think that this choice should be highlighted in the article, as clearly as possible.

Why in Figure 1 the authors state that ''Studies included in qualitative and quantitative synthesis (meta-analyses) (n = 5/7)''? .......there are not 6 articles analyzed in the end, the figure shows that there are 5 or 7, please explain

Reviewer #3: The background of the paper (Line 11-15) is about sport and the promotion of respect in the practice of sport among young people, but later in the paper, we can see that authors mention teachers (Line 40, line 45...) and physical education classes, so it is necessary to harmonize the terminology. We can see that there are six types of research included in the quantitative analysis, four were performed in Secondary Education with boys and girls between 12 and 18 years old, one in Primary Education (10-12 years old), and only one in a football club (10-12 years old). Authors cannot state that “the development of respect in the context of sport can be successfully achieved through a planned and deliberate intervention” based on only one study.

The teacher and the coach have different goals so we cannot look at these problems together in the context of sports and physical education, so (Line 68-70) refers to sportsmanship, which does not have to be relevant for physical education classes at school.

It is also important which sports are in question (individual, team sports).

Line 253-261 repetition from the method section

Line 269: substantially_different

Line 173-238 is part of the data collection but consisted of the methodology of the interventions in earlier studies. It is not very clear what is the goal of this section. It is a description of the papers and it is too long. We can see that the authors used a different methodology

Reviewer #4: General comments:

The present paper is interesting. Moreover, it is seemed to be pertinent to highlight the premise of this manuscript is a worthy one, and the authors spent a great time in the research and writing. However, there are a number of issues that need to be addressed to the manuscript prior to publication. Please note that these corrections/suggestions should not be seen as a negative against the hard work the authors have put into this manuscript.

The rationale of the present study should be further highlighted. Is the development of personal and social responsibility over the life span important? Of course, it is. Is the respect an important value related to the implementation of good behaviors in sport? Yes, the sport context follows this value and assumes it as a “rule” in and out of the field inclusively. So, could the authors further highlight the pertinence of the study?

In the method’ section, a concern of the present study was that the search strategy. Using the current keywords may not be able to include all the related papers. The authors may need some references to support the usage of the current keywords. According to the Figure 1, the initial search only included around 500 papers covering such a broad topic. Therefore, references are necessary to support these keywords. Why not include papers published before 2000? Why did the authors limit the filter of search to three languages?

Given the fact that the articles included in the systematic review presented a high level of heterogeneity, what kind of precautions did the authors take to guarantee the interpretation of the results?

Discussion

Line 347 and line 351 are similar, please rewritten.

The authors exposed two main questions to answer through the present systematic review. What did they discover regarding the effectiveness of interventions towards developing respect of boys and girls during sport?

Conclusions

The authors exposed the importance of the application model of Sport Education as well as the Personal and social Responsibility model or even the Fair Play Programme. Summing up, the authors revealed that three models will be important to promote and develop respect. But did they detect preferable situations for each one to be applied?

6. PLOS authors have the option to publish the peer review history of their article (what does this mean?). If published, this will include your full peer review and any attached files.

Reviewer #1: No

Reviewer #2: **Yes: **Badicu Georgian

Reviewer #3: **Yes: **Aleksandra Aleksić Veljković

Reviewer #4: No

---

## [Author Response · Author response to Decision Letter 0]

28 Apr 2021

REBUTTAL LETTER

Manuscript PONE-D-21-05952 [EMID:0e2927bc8a1101f0]

The development of respect in young athletes: A systematic review and meta-analysis

PLOS ONE

Reviewer 1’s comments and suggestions for authors and details of the revisions and responses

-Abstract

Line 27: total effect size should be written with its’ magnitudes

Line 29: total effect size should be written with its’ magnitudes

We have added the magnitude of the total effect size

-Introduction section

Line 65: the authors should delete (p. 15)

We have deleted (p. 15).

-Methods section

Line 118: the authors should rewrite sentence especially dates

Line 186: the authors should be attention to write capital letter throughout the article

Line 270: the authors should delete (p. 404)

We have rewritten the sentence concerning the dates

We have paid attention to capitalization throughout the article

We have deleted (p. 404)

-References

The authors should check writing style of references according to journal guidelines

We have checked the references according to journal guidelines

THANK YOU for your comments and suggestions

Reviewer 2’s comments and suggestions for authors and details of the revisions and responses

-Introduction

I think that a phrase regarding the novelty of the study would be necessary in this chapter to be introduced

We have introduced a sentence about this question:

As far as we know, the study of the effects of interventions focused on the development of respect is novel. For this reason, and considering the importance…

-Line 118 - ''articles published between 2000 16 and 2020'', I think it's a mistake of word processing, please correct it....is 2000 - 2020, or 2000, 2016, 2020???

We have rewritten the sentence concerning the dates

-Another question for the authors refers to the fact that it has been highlighted that most of the studies are from Spain. So, do you have articles from other countries or did you want to analyze especially the research in this country? it is quite strange that 5 articles are from Spain and only one from another country, please detail because this aspect is important and I think that this choice should be highlighted in the article, as clearly as possible

Regarding this question, we must stress that the search, and the subsequent analysis, did not focus on studies conducted exclusively in Spain, but these six articles were the ones that met the inclusion criteria after the search carried out in the different databases.

In this sense, we have introduced de following sentence:

It should be highlighted that, although five of the six researches had been carried out in Spain, the intention of the researchers was to include the studies carried out in any country

-Why in Figure 1 the authors state that ''Studies included in qualitative and quantitative synthesis (meta-analyses) (n = 5/7)''? .......there are not 6 articles analyzed in the end, the figure shows that there are 5 or 7, please explain

This is because for the quantitative analysis of the respect to opponents and to the rules and referees, they were analyzed by 5 studies. In addition, seven studies were reviewed with regard to respect for social conventions (see Figures 2-4). It should be remembered that one of the manuscripts included two experimental groups

THANK YOU for your comments and suggestions

Reviewer 3’s comments and suggestions for authors and details of the revisions and responses

-The background of the paper (Line 11-15) is about sport and the promotion of respect in the practice of sport among young people, but later in the paper, we can see that authors mention teachers (Line 40, line 45...) and physical education classes, so it is necessary to harmonize the terminology

In this sense, the authors of this paper consider that the practice of sports performed by young athletes can be developed, both in physical education classes, in which sports are often a subject frequently used by physical education teachers, and in sports clubs, and that in both contexts it is important to promote respect. Here we are talking about young people who practice sports, regardless of the context

-We can see that there are six types of research included in the quantitative analysis, four were performed in Secondary Education with boys and girls between 12 and 18 years old, one in Primary Education (10-12 years old), and only one in a football club (10-12 years old). Authors cannot state that “the development of respect in the context of sport can be successfully achieved through a planned and deliberate intervention” based on only one study

We have modified this statement:

Based on the study conducted, it can be concluded that the development of respect among young athletes can be successfully achieved through a planned and deliberate intervention, mainly in physical education classes

-The teacher and the coach have different goals so we cannot look at these problems together in the context of sports and physical education, so (Line 68-70) refers to sportsmanship, which does not have to be relevant for physical education classes at school

We agree with the reviewer when he says that the goals of the teacher and the coach are different. However, we believe that in teaching games, and with young people, both should have similar objectives in terms of education. For this reason, we believe that sportsmanship is relevant in both contexts

-Line 253-261 repetition from the method section

As stated by the reviewer, these lines mention the databases again, but provide new information: documents selected from each of the databases, additional studies found, and manuscripts excluded according to the inclusion criteria and studies included in the research carried out. This information complements that provided by Figure 1, and we think it’s important

-Line 269: substantially_different

We have corrected this issue

-Line 173-238 is part of the data collection but consisted of the methodology of the interventions in earlier studies. It is not very clear what is the goal of this section. It is a description of the papers and it is too long. We can see that the authors used a different methodology

The aim of this part is to describe the most important characteristics of the interventions made in the analyzed manuscripts. However, its length has been shortened as suggested by the reviewer

THANK YOU for your comments and suggestions

Reviewer 4’s comments and suggestions for authors and details of the revisions and responses

-The rationale of the present study should be further highlighted. Is the development of personal and social responsibility over the life span important? Of course, it is. Is the respect an important value related to the implementation of good behaviors in sport? Yes, the sport context follows this value and assumes it as a “rule” in and out of the field inclusively. So, could the authors further highlight the pertinence of the study?

The authors have underlined the relevance of the study:

In this way, children who participate in sport can benefit from the development of a range of personal and social skills, such as skills for relationships with peers and for personal and social responsibility, as well as pro-social behaviours such as respect. Therefore, it is pertinent to study the development of respect in young people through sport practice

-In the method’ section, a concern of the present study was that the search strategy. Using the current keywords may not be able to include all the related papers. The authors may need some references to support the usage of the current keywords. According to the Figure 1, the initial search only included around 500 papers covering such a broad topic. Therefore, references are necessary to support these keywords. Why not include papers published before 2000? Why did the authors limit the filter of search to three languages?

Dear reviewer, we think your reflections are very interesting. In this regard, it should be mentioned that for the selection and inclusion of the keywords of the search carried out, numerous documents on the subject under study were consulted, as well as the keywords of those documents. Mention should also be made of the fact that the criteria for inclusion were very demanding. This was because the goal was to obtain research that had control and experimental groups, as well as pre-test and post-test reporting results through means and standard deviations. All this in order to be able to carry out the meta-analyses

With regard to the criterion of including studies published after the year 2000, we have to say that our aim was to focus research on the twenty-first century, with the aim of analysing and including the latest research

With regard to the languages of publication, English was chosen because it is the language used by the scientific community, and Spanish and Portuguese because they are the languages we know best and are official in many countries

-Given the fact that the articles included in the systematic review presented a high level of heterogeneity, what kind of precautions did the authors take to guarantee the interpretation of the results?

As the reviewer rightly points out, the meta-analyses performed showed a high degree of heterogeneity. This leads us to interpret with caution the results obtained in the meta-analyses carried out, and this has been stated in the manuscript. In this sense, this warning has been highlighted in the discussion and conclusions sections of the paper presented, and even in its limitations. In this way, the researchers aim to ensure that the interpretation of the results obtained is properly understood by the scientific community

-Discussion

Line 347 and line 351 are similar, please rewritten

These lines have been rewritten

-The authors exposed two main questions to answer through the present systematic review. What did they discover regarding the effectiveness of interventions towards developing respect of boys and girls during sport?

A paragraph has been introduced to clarify the results found on the effectiveness of interventions to develop respect for boys and girls during sport.

On the other hand, to answer the questions of the review and meta-analysis raised above, we have to say that, overall, the interventions carried out in the studies analyzed significantly promoted respect for opponents and social conventions. In terms of the characteristics of the interventions that led to an improvement in respect, the results showed that interventions based on models of sports education and personal and social responsibility achieved a significant increase in respect

-The authors exposed the importance of the application model of Sport Education as well as the Personal and social Responsibility model or even the Fair Play Programme. Summing up, the authors revealed that three models will be important to promote and develop respect. But did they detect preferable situations for each one to be applied?

As the reviewer points out, all three models made significant improvements. The sports education model was used in secondary education, the personal and social responsibility model in secondary education and in a football club, while the fair play programme in primary education. In this respect, we have to say that, with the data provided by the studies analysed, no preferable situations were identified for the application of the different models

We believe that the question raised by the reviewer is interesting and relevant and that further research is needed to answer it. For this reason, we have added a future line of research to the article

In addition, in order to determine the most appropriate context for the implementation of a particular model, these models could be applied in different situations (primary education, secondary education, sports schools, sports clubs, etc.). This would allow us to test their effectiveness in developing respect, and to analyse in which contexts they work best

THANK YOU for your comments and suggestions

---

## [Decision Letter · Decision Letter 1]

17 May 2021

PONE-D-21-05952R1

The development of respect in young athletes: A systematic review and meta-analysis

PLOS ONE

Dear Dr. Abad Robles,

Thank you for submitting your manuscript to PLOS ONE. After careful consideration, we feel that it has merit but does not fully meet PLOS ONE’s publication criteria as it currently stands. Therefore, we invite you to submit a revised version of the manuscript that addresses the points raised during the review process.

There is only one last modification required in the manuscript prior to be accepted.

We look forward to receiving your revised manuscript.

Kind regards,

Francisco Javier Huertas-Delgado, Ph.D.

Academic Editor

PLOS ONE

Journal Requirements:

Reviewers' comments:

Reviewer's Responses to Questions

7. PLOS authors have the option to publish the peer review history of their article (what does this mean?). If published, this will include your full peer review and any attached files.

Reviewer #1: No

Reviewer #5: No

**Comments to the Author**

1. If the authors have adequately addressed your comments raised in a previous round of review and you feel that this manuscript is now acceptable for publication, you may indicate that here to bypass the “Comments to the Author” section, enter your conflict of interest statement in the “Confidential to Editor” section, and submit your "Accept" recommendation.

Reviewer #5: All comments have been addressed

2. Is the manuscript technically sound, and do the data support the conclusions?

Reviewer #5: Yes

3. Has the statistical analysis been performed appropriately and rigorously? 

Reviewer #5: Yes

4. Have the authors made all data underlying the findings in their manuscript fully available?

Reviewer #5: No

5. Is the manuscript presented in an intelligible fashion and written in standard English?

Reviewer #5: Yes

6. Review Comments to the Author

Reviewer #5: 1) In the first paragraph of Results, lines 256-257—“93 were excluded because they were systemic review …”: in Figure it shows 96 instead of 93; “… and 383, after applying the exclusion. criteria”: it’s ambiguous, please revise.

---

## [Author Response · Author response to Decision Letter 1]

18 May 2021

Reviewer #5’s comments and suggestions for authors and details of the revisions and responses

- 1) In the first paragraph of Results, lines 256-257—“93 were excluded because they were systemic review …”: in Figure it shows 96 instead of 93; “… and 383, after applying the exclusion. criteria”: it’s ambiguous, please revise.

We have made the changes indicated.

THANK YOU for your comments and suggestions.

---

## [Editor Report · Decision Letter 2]

20 May 2021

The development of respect in young athletes: A systematic review and meta-analysis

PONE-D-21-05952R2

Dear Dr. Abad Robles,

We’re pleased to inform you that your manuscript has been judged scientifically suitable for publication and will be formally accepted for publication once it meets all outstanding technical requirements.

Kind regards,

Francisco Javier Huertas-Delgado, Ph.D.

Academic Editor

PLOS ONE

---

## [Editor Report · Acceptance letter]

24 May 2021

PONE-D-21-05952R2 

The development of respect in young athletes: A systematic review and meta-analysis 

Dear Dr. Abad Robles:

I'm pleased to inform you that your manuscript has been deemed suitable for publication in PLOS ONE. Congratulations! Your manuscript is now with our production department. 

Kind regards, 

on behalf of

Dr. Francisco Javier Huertas-Delgado 

Academic Editor

PLOS ONE